

**Can Semi-Volatile Organic Aerosols Lead to Less Cloud Particles?**
**Chloe Y. Gao[1,2], Susanne E. Bauer[2], and Kostas Tsigaridis[3,2]**
[1]Department of Earth and Environmental Sciences, Columbia University, New York, NY, 10027,
USA
[2]NASA Goddard Institute for Space Studies, New York, NY, 10025, USA
[3]Center for Climate Systems Research, Columbia University, New York, NY, 10025, USA
*Correspondence to*: Kostas Tsigaridis (kostas.tsigaridis@columbia.edu)
**Abstract.** The impact of condensing organic aerosols on activated cloud number concentration is
examined in a new aerosol microphysics box model, MATRIX-VBS. The model includes the
volatility-basis set (VBS) framework coupled with the aerosol microphysical scheme MATRIX
(Multiconfiguration Aerosol TRacker of mIXing state) that resolves aerosol mass and number
concentrations and aerosol mixing state. By including the condensation of organic aerosols, the
new model produces less activated particles compared to the original model, which treats organic
aerosols as non-volatile. Parameters such as aerosol chemical composition, mass and number
concentrations, and particle sizes which affect activated cloud number concentration are
thoroughly tested via a suite of Monte-Carlo simulations. Results show that by considering semi-
volatile organics in MATRIX-VBS, there is lower activated particle number concentration, except
in cases with low cloud updrafts, in clean environment at above freezing temperatures, and in
polluted environments at high temperature (310K) and extremely low humidity conditions.
**1 Introduction**
Atmospheric aerosols influence climate mainly via two pathways: aerosol-radiation
interactions (the aerosol direct effect; Charlson et al., 1992) which affect the Earth's radiative
energy balance by absorbing and scattering terrestrial and solar radiation, and aerosol-cloud
interactions (the aerosol indirect effect; Twomey, 1974; Albrecht, 1989) which affect cloud
microphysics by activating and serving as seeds for cloud formation (Myhre et al., 2013; Seinfeld
and Pandis, 2016). Aerosol activation as cloud condensation nuclei (CNN) is critical to the
evolution and microphysics of clouds (Reutter et al., 2009). However, the relationship between
aerosol mixing state and cloud microphysical properties remain a large uncertainty in aerosol-



cloud interactions (Ghan et al., 1998; McFiggans et al., 2006; Ervens et al., 2007; Gibson et al.,
2007; Medina et al., 2007; Cubison et al., 2008; Anttila, 2010).
Climate models calculate cloud droplet number concentration (CDNC) using aerosol
activation schemes, whose main governing parameters include aerosol number, size,
hygroscopicity, updraft velocity, as well as critical supersaturation. Physically-based aerosol
activation schemes (e.g. Abdul-Razzak and Ghan, 2000; Fountoukis and Nenes, 2005; Ming et al.,
2006; Shipway and Abel, 2010) are commonly used in global climate models for fast diagnostics
of nucleation and to estimate the aerosol indirect effect in long-term climate simulations (Ghan,
2011). Several studies examined the relationship between the fore-mentioned parameters and how
they play together to activate particles. Ghan et al. (1998) examined sea salt's influence on sulfate
particle activation and introduced the competition effect. Since all CCN have to compete for
available water vapor in order to activate, the competition limits the maximum supersaturation in
in-cloud updrafts (Storelvmo et al., 2006). Ghan et al. (1998) concluded that activated number
concentration increases with increasing sea salt when sulfate is low and updraft is strong, and it
decreases when sulfate is high and updraft is weak, because maximum supersaturation is reduced.
Another study (Reutter et al. 2009) explored how much CDNC depend on updraft velocity, size
distribution and hygroscopicity. They found that size distribution played a greater role than particle
hygroscopicity on CDNC and discovered different CCN activation and cloud droplet formation
regimes, which are determined by aerosol number concentration and updraft velocity.
Semi-volatile organic aerosols contribute significantly to the growth of particles to CCN
sizes (Yu, 2011). More notably, as aerosol size increases, the range of organic volatilities involved
in aerosol growth increases (Pierce et al., 2011; Yu, 2011). The inclusion of semi-volatile organics
in models modifies CCN formation rates (Petters et al., 2006, Riipenen et al., 2011; Scott et al.,
2015) as well as hygroscopicity (Petters and Kreidenweis, 2007), in addition to bulk aerosol mass,
size distribution and composition. By adding semi-volatile organic partitioning to our existing
microphysics model MATRIX (Multiconfiguration Aerosol TRacker of mIXing state; Bauer et al.,
2008), which resolves aerosol mixing state, we were able to examine how they change bulk aerosol
mass, size distribution and composition. However, the effects of semi-volatile organic partitioning
combined with aerosol mixing state on particle activation remain unexplored.
In our previous work, we demonstrated that including semi-volatile organics would lead to
higher aerosol number concentration and smaller particles (Gao et al., 2017). As was the case for





the original aerosol microphysics model MATRIX, our further-developed box model MATRIX-
VBS (Gao et al., 2017) follows the same multi-modal aerosol activation approach by Abdul-
Razzak and Ghan (2000). The activation parameterization accounts for aerosol size distribution,
composition, mixing state, and in-cloud updraft velocity. Curious about the change in activation
with the newly-present semi-volatile organics and the governing parameters influencing it, we
investigated the difference in activated number concentration in two box model set ups: MATRIX
(Bauer et al., 2008) and MATRIX-VBS (Gao et al., 2017).
**2 Methods**
2.1 Model Description

71        MATRIX-VBS (Gao et al., 2017) is an aerosol microphysics model that includes organic

aerosol volatility in its calculations. It was developed by implementing VBS (volatility-basis set;
Donahue et al., 2006) in the aerosol microphysics model MATRIX (Bauer et al., 2008), which is
a box model that is also used in the NASA GISS ModelE Earth System Model (Bauer et al., 2008,
2012; Schmidt et al., 2014). Since the publication of Gao et al., 2017, which included organic
condensation on fine mode aerosols, we further developed the model which now allows semi-
volatile organics in the system to condense on coarse mode dust and sea salt as well. We have also
included nitrate radicals as an oxidant for organics in addition to the hydroxyl radical that was used
in the original VBS scheme, even though it is a very minor oxidation pathway in the model (rate
constant for the oxidation by $NO_3^{\bullet}$ is $1*10^{-13}$ $cm^3$ molecules$^{-1}$ s$^{-1}$; Atkinson, 1997). As previously
stated, we use Abdul-Razzak and Ghan (2000) activation parameterization, which calculates the
activated particle number concentration depending on chemically-resolved number concentrations
using Köhler Theory. The hygroscopicity parameters κ for each aerosol species presented in Table
1 were calculated from their solubility fraction. For organics, we assumed a linear increase of
solubility with decreasing volatility (Jimenez et al., 2009).
2.2 Simulations

87        A Monte-Carlo analysis with a range of chemical and meteorological conditions (Table 2)

was performed, to pinpoint which processes affect organics and the mixed aerosol population in
general the most. Since global models need to resolve a wide range of conditions, from very clean
to very polluted and for a wealth of meteorological conditions, we simulated 630 possible





atmospheric scenarios on Earth across the whole parameter space, e.g. temperature, relative
humidity, latitude, emissions levels and updraft velocity, for 120 hours (5 days) simulations with
no deposition and dilution. Three types of environmental conditions were simulated: clean,
moderate and polluted, as defined by different levels of emissions which were determined using a
probability distribution of the gridded emission fields in GISS ModelE for January present-day
conditions. During this development phase, biogenic secondary organic aerosols from terpenes
oxidation in MATRIX-VBS are treated as nonvolatile, while only the anthropogenic aerosols are
treated as semi-volatile.
**3 Results and discussion**
We found that activated number concentration is lower for most cases in the MATRIX-
VBS model, which considers semi-volatile organic aerosols, as compared to the MATRIX model.
However, under low updrafts, in clean environment at above freezing temperatures, and in polluted
environments at high temperature (310K) and extremely low humidity conditions (0% RH) during
aerosol formation, activated number concentration is higher in MATRIX-VBS than in MATRIX.
As an example, the activated number concentration for a case with temperature at 290˚K,
relative humidity at 40%, medium emission levels and an updraft of 0.5 m/s at 30°N latitude is
shown in Figure 1 for the two models. Mixing states of aerosols in MATRIX and MATRIX-VBS
are represented as aerosol populations, which all contain $SO_4$, $NO_3$, $NH_4$ and $H_2O$, in addition to
the species that define the populations (Bauer et al., 2008, 2013). The four most dominant aerosol
populations for the activated number concentration in MATRIX are ACC ($SO_4$, $NO_3$, $NH_4$), OCS
(organics, $SO_4$, $NO_3$, $NH_4$), BOC (black carbon, organic carbon, $SO_4$, $NO_3$, $NH_4$) and BCS (black
carbon, $SO_4$, $NO_3$, $NH_4$). Only two dominant populations are calculated in MATRIX-VBS, OCS
and BOC, as in Gao et al., 2017, since OCC evaporates and re-condenses on all particles, based on
their calculated surface area and mass concentration. Since OCS and BOC have the largest surface
area, they are calculated to have the strongest growth via organics condensation. Additionally, the
competition between sulfate, organics and black carbon, determines the loss of ACC and the
formation of BCS: OCC coagulates with ACC to form OCS, and this coagulation increases in
MATRIX-VBS due to smaller OCC particles; therefore, there are less ACC particles left to
coagulate with black carbon to form BCS. At the end of the 5-day simulation (Figure 1), MATRIX-




VBS has approximately a total of 30 activated particles/cm$^3$, whereas MATRIX has approximately
60 activated particles/cm$^3$ under the same conditions.
Figure 2 shows a more comprehensive look across all temperature and relative humidity
scenarios studied. The results show that for most scenarios, MATRIX-VBS has lower (red circles)
activated number concentration compared to MATRIX. However, some rare cases show the
opposite behavior. These are for above freezing temperatures in the low emission level under low
updraft (top left) scenarios, high temperature (310K) and extremely low humidity (0% RH) in the
medium emission level under low updraft (middle left) scenarios, as well as the high emission
level under low (bottom left) and medium (bottom middle) updraft scenarios.  Across all scenarios,
the changes in activated number concentration between MATRIX-VBS and MATRIX range from
a -56% to +31% (Table 3). The range of the difference becomes more significant as emission levels
increase, yet less significant as updraft velocity increases. Within most emission level-updraft
velocity scenarios, as temperature increases, the fractional change in activated number
concentration between the two models decreases. Also within most emission level-updraft velocity
scenarios (Figure 3, Table 4), as temperature increases, there are less activated particles in
MATRIX. We also observed the same behavior in MATRIX-VBS, higher temperature, less
activated particles.
In order to understand the cause of the difference in activation, we traced back to the key
difference between the two models: partitioning of organics. The inclusion of organics partitioning
leads to changes in aerosol mixing state and size distribution, as discussed in Gao et al. (2017).
Therefore, the change in activated number concentration could only be caused by changes in mass
concentration, number concentration and particle size. Since we use the Abdul-Razzak and Ghan
(2000) parameterization, and the activated number concentration is only a function of number
concentration and dry particle diameter.
As was the case in Gao et al., (2017), MATRIX-VBS has higher aerosol number
concentration (Figure 4 left) but smaller particles (Figure 4 right) compared to MATRIX in the
case presented in Figure 1. At first we expected that smaller particles would less likely activate, so
we performed a simple sensitivity test to confirm it. By changing dry particle diameter of the
particles in the activation scheme, the decreasing dry particle diameter indeed led to lower
activated number concentration.  However, a second sensitivity test with changing only number



concentration showed that higher number concentration would actually lead to lower activated
number concentration as well.

152        In the Abdul-Razzak and Ghan (2000) scheme, increasing number concentration decreases

critical supersaturation, and lower critical supersaturation leads to higher minimum dry particle
radius that is able to activate. Therefore, activation is suppressed, since less particles exceed the
threshold radius. The activated number concentration is calculated from the activation fraction and
the number concentration. When the fraction is greater than the increase in number concentration,
lower activated number concentration is achieved, as shown here.

158        As mentioned previously, within most of the scenarios, there is a decrease in fractional

change as temperature increases, while both models experience decrease in activated number
concentration with increased temperature. This means the decrease in activated number
concentration for MATRIX-VBS is not as significant as that for MATRIX. There are two factors
that contribute to such change. First, the heat and moisture diffusion term is dependent on
temperature in the activation scheme (Abdul-Razzak and Ghan, 2000). Second, volatility of
organics is temperature dependent. In MATRIX-VBS, when organic volatility is considered, the
change is dampened. In other words, its number of activated particles is less sensitive to
temperature change as compared to MATRIX, leading to what we see in the circle plots that the
greater change at lower temperatures.

168        The length of day and season changes the duration and intensity of gas phase oxidation of

semi-volatile gases, which is why we also looked at aerosol evolution driven by photochemistry
at different latitudes. Since the model uses January emissions, different seasons are simulated at
the different hemispheres, while different day lengths are simulated at higher latitudes of the
southern hemisphere compared to tropical and high latitude northern hemisphere ones. As we
inspected results across latitudes in the two hemispheres, we found varying activated number
concentration in MATRIX-VBS compared to MATRIX and observed no evident trend. Such
inconclusive and complex results may be due to gas-phase chemistry and photochemical ageing
of semi-volatile organic vapors, which would require further examination in a separate dedicated
study.



## 4 Conclusions

With the inclusion of organic partitioning in an aerosol microphysics model, activated aerosol number concentration is decreased under most temperature and relative humidity conditions, except when under low updrafts, in clean environments at most temperatures and relative humidities, and in polluted environments at high temperatures and extremely low humidity conditions. Such changes are due to increased aerosol number concentration and smaller particles in the new model, as well as how number concentration and size are calculated in the chosen aerosol activation scheme, which determines how many particles are activated. Additionally, the temperature dependence of activated number concentration is decreased for most scenarios.

The simulations in this study, however comprehensive, are still highly idealized. In fact, Topping et al. (2013) showed that co-condensing organics lead to enhanced cloud droplet number concentration, which seems to contradict our results. However, it is important to note that our study is performed in a box model that does not resolve cloud physics. Activated number concentration is a precursor for CDNC, whose actual numbers will depend on the cloud microphysical calculation, which is not part of this study. We will investigate the effects of condensing organics in a global climate model in the future. The results presented here implicate that in the new model, most areas on Earth would experience less CCN on a typical day, but clean environments with above freezing temperatures, or polluted environments on an extremely dry and hot day, would form more CCN under low updraft velocity conditions, as compared to the old model. We expect that implementing the improved box model in the global scale that includes a two moment cloud microphysical scheme (Morrison and Gettelman, 2008; Gettelman and Morrison, 2015) would more accurately represent aerosol-cloud interactions, which will be our focus on a follow up study. Thus it would offer us valuable insights on how the addition of organic partitioning would change cloud activation in the global atmosphere and its implications for climate.

**Acknowledgements.** We thank the NASA Earth and Space Science Fellowship Program (17-EARTH17F-85) and the NASA Modeling, Analysis, and Prediction Program for supporting Chloe Y. Gao's graduate study, as well as the NASA Atmospheric Composition Modeling and Analysis





Program (NNX15AE36G) for supporting Dr. Susanne E. Bauer and Dr. Kostas Tsigaridis. We also
thank Dr. Steven Ghan, Dr. Hyunho Lee and Dr. Ann Fridlind for sharing their insights with us.
The GISS ModelE Earth system model is publicly available. The box model code used here is
available upon request and will be publicly available in the future as part of GISS ModelE. The
data from all model simulations will be available upon request.
The authors declare that they have no conflict of interest.



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



**Table 1. Hygroscopicity κ used for each organic aerosol volatility bin.**

|  | $\log_{10}C^*$ [μg m$^{-3}$] | soluble fraction [%] | κ |
|---|---|---|---|
| Sulfate | / | 100 | 0.507 |
| Black carbon | / | 0 | $5 \cdot 10^{-7}$ |
| Non-volatile organic carbon | / | 78 | 0.141 |
| | -2 | 100 | 0.180 |
| | -1 | 87.5 | 0.158 |
| | 0 | 75 | 0.135 |
| | 1 | 62.5 | 0.113 |
| Semi-volatile organic carbon | 2 | 50 | 0.090 |
| | 3 | 37.5 | 0.068 |
| | 4 | 25 | 0.045 |
| | 5 | 12.5 | 0.023 |
| | 6 | 0 | 0.000 |
| Dust | / | 13 | 0.14 |
| Sea salt | / | 100 | 1.335 |

**Table 2. Parameters used in the Monte-Carlo simulations.**

| Parameter | | Range |
|---|---|---|
| T [K] | | 270, 280, 290, 300, 310 |
| RH [%] | | 0.1, 20, 40, 60, 80, 100 |
| Latitude | | 0, 30N/S, 60N/S, 90N/S |
| Updraft velocity [m/s] | | 0.5, 1, 2 |
| Emissions of aerosols [μg/m$^3$/s] | Sulfate (SO$_2$ in molecules/cm$^3$) | $10^5$, $10^6$, $5 \cdot 10^6$ |
| | Primary organics | $5 \cdot 10^{-6}$, $5 \cdot 10^{-5}$, $5 \cdot 10^{-4}$ |
| | Nonvolatile biogenic organics from terpene source | $1 \cdot 10^{-8}$, $5 \cdot 10^{-6}$, $1 \cdot 10^{-5}$ |
| | Black Carbon | $10^{-6}$, $10^{-5}$, $10^{-4}$ |
| Emissions of gases [molecules/cm$^3$] | VOCs (in sets) — Alkenes | $5 \cdot 10^2$, $5 \cdot 10^3$, $5 \cdot 10^4$ |
| | Paraffin | $5 \cdot 10^3$, $10^4$, $5 \cdot 10^4$ |
| | Terpenes | $10^4$, $10^5$, $10^6$ |
| | Isoprene | $10^4$, $10^5$, $50^6$ |
| | NO$_x$ | $10^5$, $10^6$, $10^7$ |






**Table 3. Minimum and maximum of fractional change in average activated number concentration over the last 24 hours between the two models with low, medium and high level emissions at updraft velocities of 0.5, 1 and 2 m/s.**

| | Fractional change in activated number concentration | | | | | |
|---|---|---|---|---|---|---|
| Updraft velocity (m/s) | 0.5 | | 1 | | 2 | |
| | min | max | min | max | min | max |
| Low emission level | -9% | +21% | -16% | +2% | -14% | +5% |
| Medium emission level | -51% | +14% | -42% | -5% | -36% | -13% |
| High emission level | -56% | +31% | -48% | +9% | -43% | -9% |

**Table 4. Minimum and maximum of average activated number concentration over the last 24 hours of MATRIX and MATRIX-VBS with low, medium and high level emissions at updraft velocities of 0.5, 1 and 2 m/s.**

| | | Activated number concentration | | | | | |
|---|---|---|---|---|---|---|---|
| Updraft velocity (m/s) | | 0.5 | | 1 | | 2 | |
| | | min | max | min | max | min | max |
| Low emission level | MATRIX | 23 | 305 | 351 | 1160 | 963 | 2799 |
| | MATRIX-VBS | 24 | 283 | 338 | 1026 | 887 | 2473 |
| Medium emission level | MATRIX | 19 | 152 | 359 | 1233 | 1476 | 3711 |
| | MATRIX-VBS | 16 | 139 | 304 | 884 | 1021 | 2498 |
| High emission level | MATRIX | 3 | 60 | 199 | 1280 | 1925 | 5703 |
| | MATRIX-VBS | 3 | 63 | 185 | 1150 | 1677 | 4142 |





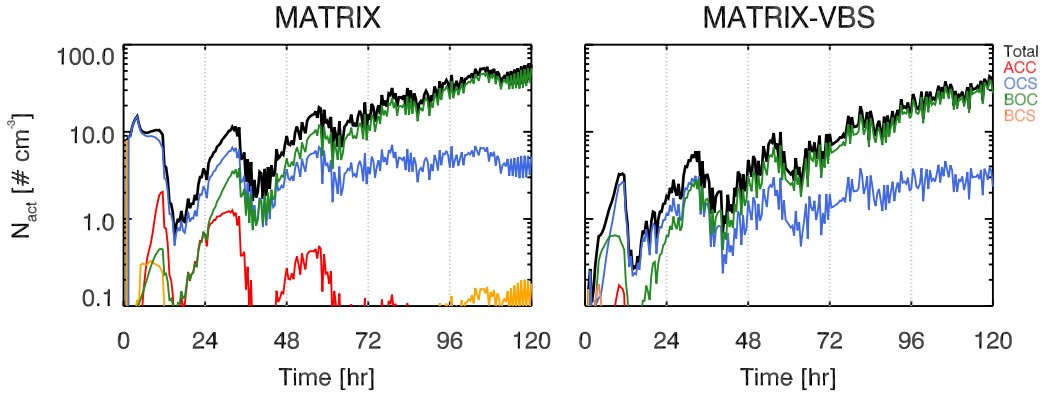


**Figure 1. Activated number concentration of aerosol populations (see main text for details) for MATRIX (left) and MATRIX-VBS (right) for 290 K and 40% RH at 30°N latitude with medium emission levels and 0.5 m/s updraft velocity.**






**Figure 2. Fractional change of average activated number concentration (size and color of the circles) over the last 24 hours of a 5-day simulation between the two models with low (top row), medium (middle row) and high (bottom row) level emissions at updraft velocities of 0.5 (left column), 1 (middle column) and 2 (right column) m/s.**

413





414
**Figure 3. Average activated number concentration (circle size) during the last 24 hours of a 5-day simulation in MATRIX and MATRIX-VBS with low (top row), medium (middle row) and high (bottom row) emission levels at updraft velocities of 0.5 (left column), 1 (middle column) and 2 (right column) m/s. Note difference in scales per column.**



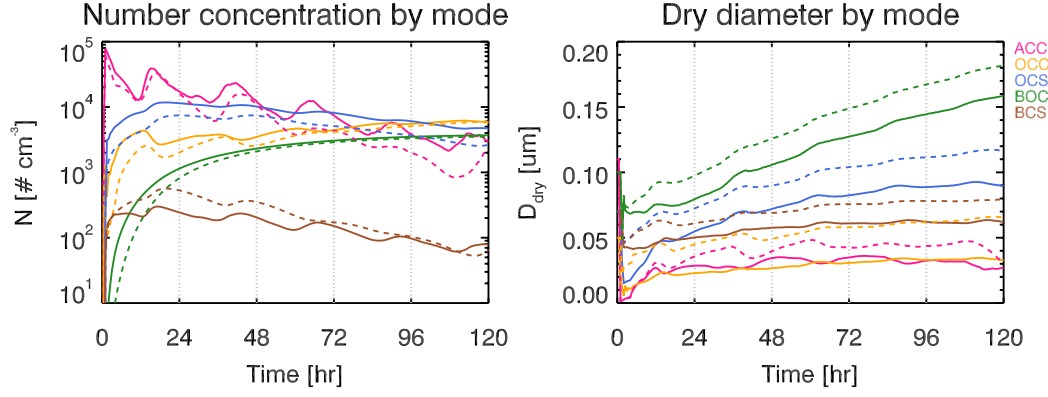

419

**Figure 4. Number concentration (left column) and dry particle diameter (right column) by mode (color lines) for MATRIX (dashed lines) and MATRIX-VBS (solid lines) for the experiments with the same conditions as Figure 1.**