# Peer review of "Chloe Y. Gao1,2, Susanne E. Bauer2, and Kostas Tsigaridis3,2"

_Atmospheric Chemistry and Physics, 2018_

## Referee Comment (RC1) · Anonymous Referee #1 · 1 May 2018

Review of "Can Semi-Volatile Organic Aerosols Lead to Less Cloud Particles?"

This paper investigates the sensitivity of cloud droplet activation, using the Abdul-Razzak et al. (2000) parametrisation, to the aerosol chemical composition, mass and number concentrations, and particle size. The main finding is that the simulations suggest that semi-volatile compounds almost always lead to fewer cloud droplets than without semi-volatiles. Whilst I agree this area is worthy of further investigations, I believe the main finding is an artefact of the parameterisation used – the Abdul-Razzak et al. (2000) scheme and perhaps also in the way it is implemented in the model

There are a few statements that lead me to conclude the authors might not be treating activation in the correct way. For example, on line 141 the authors write "the activated number concentration is only a function of number concentration and dry particle diameter". This is not true: in addition to environmental parameters such as temperature and pressure, activation is also a function of the geometric standard deviation (see equation 15 of ARG, 2000), and the aerosol chemistry (see equation 3 of ARG, 2000).

Of the parameters above the geometric standard deviation is an important parameter for cloud drop activation. If the distribution is narrow (small geometric standard deviation) then the competition effect will be small and more particles will activate at once. If the aerosol size distribution is broad / geometric standard deviation is large one tends to find that fewer particles activate. Connolly et al. (2014) showed that for single aerosol modes it was necessary to shift the geometric standard deviation to smaller values in order to predict activated fractions more accurately – see their equation 15. Crooks et al. (2018) have now extended this result to multiple modes.

Please clarify whether this shifting done in the current manuscript.

On line 152 the authors mention that, in the Abdul-Razzak et al. (2000) scheme, increasing number concentration decreases the ambient supersaturation, which reduces the number activated, therefore suppressing activation. This argument is slightly circular though since to reduce the ambient supersaturation more particles must have been activated.

We know that the Abdul-Razzak et al. parameterisation does not always predict the correct response to inputs. Connolly et al. (2014b) showed a comparison between the Abdul-Razzak et al. (2000) and the Fountoukis and Nenes (2005) parameterisations. Their Figure 3(a-d) is reproduced in Figure 1, below. Figure 1a shows how increasing the total aerosol mass (by increasing the aerosol particle number concentration, the x-axis) eventually leads to less particles being activated in the Abdul-Razzak et al. parameterisation. Such a reduction is not seen in the Fountoukis and Nenes (2005) parameterisation (see Figure 1d).

[Figure]

**Figure 1. shows the activated fraction when adding NaCl particles to the aerosol population with a total mass loading indicated by the x-axis. Colours refer to different modal diameters of the NaCl particle size distribution (see Connolly et al, 2014b for full details). (a) is for the Abdul-Razzak et al. parameterisation; (b) is for the Fountoukis and Nenes parameterisation.**

The results in the presented manuscript, that fewer particles are activated with semi-volatiles for higher updrafts, are also in contrast to Connolly et al. (2014), which found ARG at low updraft speeds activated fewer particles with semi-volatiles switched on; (see Figure 6 of Connolly et al, 2014). I suspect the reason for this contrast between the two studies is that the Abdul-Razzak et al. parameterisation gives a different response when multiple aerosol modes are used, as has been shown by Connolly et al. (2014b). Indeed results by Simpson et al (2014), which are reproduced in Figure 2, indicate that this is the case. The ARG parameterisation results are indicated by the '+' symbols and it is shown that ARG is further below the 1 : 1 line when the updrafts are high (red) vs when the updraft is low (blue).

[Figure]

Figure 2. Results from Simpson et al. (2014). Using a biomodal aerosol size distribution. Symbols coloured by updraft velocity (m s-1).

On line 126 there is mention of simulations looking at activation at extremely low humidity. How relevant are these simulations, given that activation would not occur at low RH anyway?

The statement on line 188 about the discrepancy between the results presented and those of Topping et al. (2013) seems to indicate that the differences are because Topping et al. (2013) resolved more physics than the cloud drop activation process. In fact this is not really true. The Topping et al. study only considered condensation until the point of cloud drop activation.

Unfortunately, because of these shortcomings I feel like the conclusions drawn about most areas on earth experiencing less CCN that currently thought, except the more polluted & dry areas, are all dependent on the parameterisation used and its implementation.

**References**

Connolly, P. J., Topping, D. O., Malavelle, F., and McFiggans, G. 2014. A parameterisation for the activation of cloud drops including the effects of semi-volatile organics, Atmos. Chem. Phys., 14, 2289-2302, https://doi.org/10.5194/acp-14-2289-2014

Connolly PJ, McFiggans GB, Wood R, Tsiamis A. 2014b Factors determining the most efficient spray distribution for marine cloud brightening. *Phil. Trans. R. Soc. A* **372**: 20140056. http://dx.doi.org/10.1098/rsta.2014.0056

Simpson, E., Connolly, P., and McFiggans, G., 2014. An investigation into the performance of four cloud droplet activation parameterisations, Geosci. Model Dev., 7, 1535-1542, https://doi.org/10.5194/gmd-7-1535-2014

---

## Short Comment (SC1) · 18 May 2018

Reply to the review of "Can Semi-Volatile Organic Aerosols Lead to Less Cloud Particles?"

We would like to thank the reviewer for their careful reading and constructive comments. Please find below our answers to all points raised. The original reviewer's comments are in black font, and our replies are in blue. Changes in the text are *italicized*.

1. This paper investigates the sensitivity of cloud droplet activation, using the Abdul-Razzak et al. (2000) parameterisation, to the aerosol chemical composition, mass and number concentrations, and particle size. The main finding is that the simulations suggest that semi-volatile compounds almost always lead to fewer cloud droplets than without semi-volatiles. Whilst I agree this area is worthy of further investigations, I believe the main finding is an artefact of the parameterisation used – the Abdul-Razzak et al. (2000) scheme and perhaps also in the way it is implemented in the model.

The Abdul-Razzak and Ghan (2000) scheme is widely used in global climate models and in MATRIX (Bauer et al., 2008), which is the original version of the model that we compare the new model against. We are not aware of an artefact being reported in the literature. We also checked the implementation, and found no obvious problems with it. As a matter of fact, the Connolly et al. (2014) and Crooks et al. (2017) papers mentioned by the reviewer as more accurate in terms of aerosol activation calculations, use that same ARG scheme, only with different input parameters, appropriate for each individual study.

2. There are a few statements that lead me to conclude the authors might not be treating activation in the correct way. For example, on line 141 the authors write "the activated number concentration is only a function of number concentration and dry particle diameter". This is not true: in addition to environmental parameters such as temperature and pressure, activation is also a function of the geometric standard deviation (see equation 15 of ARG, 2000), and the aerosol chemistry (see equation 3 of ARG, 2000).

Thanks for pointing this out. This was worded imprecisely and may have delivered the wrong message, the activated number concentration is not "only" a function of number concentration and dry particle diameter in the ARG scheme. We used "only" because in our model the geometric standard deviation is constant for each aerosol population. Aerosol composition, which is calculated explicitly per population by the aerosol microphysics of the model, and hygroscopicity, are indeed taken into account in our model. We mention hygroscopicity in the submitted manuscript in lines 80-85: "*As*

*previously stated, we use Abdul-Razzak and Ghan (2000) activation parameterization, which calculates the activated particle number concentration depending on chemically-resolved number concentrations using Köhler Theory. The hygroscopicity parameters κ for each aerosol species presented in Table 1 were calculated from their solubility fraction. For organics, we assumed a linear increase of solubility with decreasing volatility (Jimenez et al., 2009).*" Therefore, we considered them not changing variables per aerosol component, but due to size distribution evolution the hygroscopicity of aerosol populations evolves with time and is taken into account in our calculations.

To make this point clearer in the manuscript, this sentence has been revised as follows: "*Since we use the Abdul-Razzak and Ghan (2000) parameterization, the activated number concentration is mainly a function of number concentration and dry particle diameter in our model setting. The parameterization is also a function of geometric standard deviation, which is constant per population in our model as it did in MATRIX (Bauer et al., 2008), as well as a function of aerosol composition and hygroscopicity, as mentioned in the model description, for which we assume a linear increase of solubility with decreasing volatility. The hygroscopicity of the aerosol populations changes with time, as the internal mixing of aerosol populations is altered by aerosol microphysics.* "

3. Of the parameters above the geometric standard deviation is an important parameter for cloud drop activation. If the distribution is narrow (small geometric standard deviation) then the competition effect will be small and more particles will activate at once. If the aerosol size distribution is broad / geometric standard deviation is large one tends to find that fewer particles activate. Connolly et al. (2014) showed that for single aerosol modes it was necessary to shift the geometric standard deviation to smaller values in order to predict activated fractions more accurately – see their equation 15. Crooks et al. (2018) have now extended this result to multiple modes.

Please clarify whether this shifting done in the current manuscript.

We have not implemented geometric standard deviation shifting, and it remains constant both in MATRIX and other aerosol microphysics modules used by other models. MATRIX is based on the quadrature methods of moments and in a future version we might be able to treat higher moments of the size distribution, but at this point this version is not developed. But we will keep in mind the sensitivity of activation towards the standard deviation, and will perform future sensitivity experiments for that quantity.

We have included the following in our conclusions section:

*"In our study, the geometric standard deviation remained constant per aerosol population. However, it is worth exploring in the future to use reduced geometric standard deviation in our calculations to directly compare with values used by Connolly et al. (2014) and Crooks et al. (2017)."*

4. On line 152 the authors mention that, in the Abdul-Razzak et al. (2000) scheme, increasing number concentration decreases the ambient supersaturation, which reduces the number activated, therefore suppressing activation. This argument is slightly circular though since to reduce the ambient supersaturation more particles must have been activated.

It would have been circular if we were saying "increasing activated number concentration decreases ambient supersaturation" but we are saying "increasing number concentration decreases the ambient supersaturation."

5. We know that the Abdul-Razzak et al. parameterisation does not always predict the correct response to inputs. Connolly et al. (2014b) showed a comparison between the Abdul-Razzak et al. (2000) and the Fountoukis and Nenes (2005) parameterisations. Their Figure 3(a-d) is reproduced in Figure 1, below. Figure 1a shows how increasing the total aerosol mass (by increasing the aerosol particle number concentration, the x-axis) eventually leads to less particles being activated in the Abdul-Razzak et al. parameterisation. Such a reduction is not seen in the Fountoukis and Nenes (2005) parameterisation (see Figure 1d).

Figure 1. shows the activated fraction when adding NaCl particles to the aerosol population with a total mass loading indicated by the x-axis. Colours refer to different modal diameters of the NaCl particle size distribution (see Connolly et al, 2014b for full details). (a) is for the Abdul-Razzak et al. parameterisation; (b) is for the Fountoukis and Nenes parameterisation.

As mentioned by the reviewer, *"Figure 1a shows how increasing the total aerosol mass (by increasing the aerosol particle number concentration, the x-axis) eventually leads to less particles being activated in the Abdul-Razzak et al. parameterisation."* This is essentially what we have also found.
We included the following into our conclusions: *"In fact, in a comparison study, Ghan et al. (2011) found that the Abdul-Razzak and Ghan (2000) scheme tend to have lower activation fractions and droplet concentrations compared to the Fountoukis and Nenes (2005) activation scheme."*

Connolly et al. (2014b)'s result that the Fountoukis and Nenes parameterization does not give the same result as the ARG parameterization is sound, but not within the scope of our study, since it is not in our goal to perform an activation schemes comparison, but to compare two models with the same activation scheme but different organic aerosol volatility treatment. The results of our paper are based on the changes volatile organics cause in aerosol microphysical properties, changes in mass, number, size, composition, and how this affects activation. We are not testing the activation schemes themselves, as we want to be consistent with parameterizations within the GISS GCM, the model this new scheme is currently being implemented in.

6. The results in the presented manuscript, that fewer particles are activated with semi-volatiles for higher updrafts, are also in contrast to Connolly et al. (2014), which found ARG at low updraft speeds activated fewer particles with semivolatiles switched on; (see Figure 6 of Connolly et al, 2014). I suspect the reason for this contrast between the two studies is that the Abdul-Razzak et al. parameterisation gives a different response when multiple aerosol modes are used, as has been shown by Connolly et al. (2014b). Indeed results by Simpson et al (2014), which are reproduced in Figure 2, indicate that this is the case. The ARG parameterisation results are indicated by the '+' symbols and it is shown that ARG is further below the 1 : 1 line when the updrafts are high (red) vs when the updraft is low (blue).

Figure 2. Results from Simpson et al. (2014). Using a bimodal aerosol size distribution. Symbols coloured by updraft velocity (m s-1).

We have included these differences in our conclusions section as follows:
"*Our conclusion that fewer particles are activated at higher updrafts is in contrast to Connolly et al. (2014), who found that fewer particles activated at low updrafts, using a different geometric standard deviation in the same parameterization of aerosol activation as the one we use. Such a difference can be due to the fact that the Abdul-Razzak and Ghan (2000) activation parameterization produces a different response when multiple modes are used, as shown by Connolly et al. (2014b) and Simpson et al. (2014).*"

7. On line 126 there is mention of simulations looking at activation at extremely low humidity. How relevant are these simulations, given that activation would not occur at low RH anyway?

Admittedly the explanation in the manuscript was too brief regarding the RH values below 100%. We calculate aerosol formation and growth via MATRIX-VBS microphysics outside of clouds, where RH can have any value, even significantly below 100%. Then we calculate aerosol activation, assuming that the air mass containing those aerosols started rising with a given updraft velocity, and supersaturation conditions developed, emulating cloud formation.

We added a sentence following the line referred by the reviewer to explain this: "*Note that low RH values do not mean that these correspond to cloud conditions. Aerosols form outside of clouds in our model, where RH can be very low. Activation though will occur after aerosol formation, when an air parcel starts rising with a given updraft velocity, in which air parcel supersaturation will develop and will cause aerosol activation.*"

8. The statement on line 188 about the discrepancy between the results presented and those of Topping et al. (2013) seems to indicate that the differences are because Topping et al. (2013) resolved more physics than the cloud drop activation process. In fact this is not really true. The Topping et al. study only considered condensation until the point of cloud drop activation.

Topping et al. (2013) uses an adiabatic cloud parcel model to study cloud droplet formation, which does resolve some cloud physics, contrary to our box model, which does not. This is clearly stated in the abstract and throughout the Topping et al. (2013) paper, that describes cloud droplet formation and growth as an air parcel rises, which goes beyond just activation. That study, however, same as ours, does not handle autoconversion, but it does track the air mass as it rises and cools, which leads to additional condensation of organic vapors and water due to the temperature decline, leading to cloud droplet growth due to additional water uptake, which our model does not resolve. What our model can provide is the initial activation of an aerosol population when it first enters a supersaturated air mass with a given updraft velocity, and not the evolution of cloud droplets inside the rising plume.

We modified our statement which now reads: "*Topping et al. (2013) showed that co-condensing organics lead to enhanced cloud droplet number concentration, which seems to contradict our results. However, it is important to note that contrary to Topping et al. (2013), our study is performed in a box model that does not resolve cloud droplet growth as the air mass rises and cools, which leads to additional condensation of organic vapors and water due to the temperature decline, and contributes to both cloud*

*droplet growth due to additional water uptake and enhanced activation during convection.* "

9. Unfortunately, because of these shortcomings I feel like the conclusions drawn about most areas on earth experiencing less CCN that currently thought, except the more polluted & dry areas, are all dependent on the parameterisation used and its implementation.

We would like to reiterate that our goal is to compare two versions of an aerosol microphysics model with different treatment of organic aerosols and the same activation scheme. Our results do not imply that the Earth has less CCN than currently thought, as the reviewer states; instead, they imply that if in a model semi-volatile organics will be simulated together with aerosol microphysics, a general decrease is to be expected.

We included this statement in our conclusion:
"*We would like to emphasize that our results do not imply that the Earth has less CCN than currently thought, instead, they imply that if in a model semi-volatile organics will be simulated together with aerosol microphysics, a general decrease is to be expected during initial cloud formation.* "

As mentioned before, we added semi-volatile organics and they changed number and size of particles, which affected the activated number concentration via the parameterization. See lines 183-185 of the submitted manuscript: "*Such changes are due to increased aerosol number concentration and smaller particles in the new model, as well as how number concentration and size are calculated in the chosen aerosol activation scheme, which determines how many particles are activated.*"

References

Abdul-Razzak, H. and Ghan, S. J.: A parameterization of aerosol activation: 2. Multiple aerosol types, J. Geophys. Res. Atmospheres, 105(D5), 6837–6844, doi:10.1029/1999JD901161, 2000.

Bauer, S. E., Wright, D. L., Koch, D., Lewis, E. R., McGraw, R., Chang, L.-S., Schwartz, S. E., and Ruedy, R.: MATRIX (Multiconfiguration Aerosol TRacker of mIXing state): an aerosol microphysical module for global atmospheric models, Atmos. Chem. Phys., 8, 6003–6035, doi:10.5194/acp-8-6003- 2008, 2008.

Connolly, P. J., Topping, D. O., Malavelle, F., and McFiggans, G.: A parameterisation for the activation of cloud drops including the effects of semivolatile organics, Atmos. Chem. Phys., 14, 2289-2302, https://doi.org/10.5194/acp-14-2289-2014, 2014.

Connolly PJ, McFiggans GB, Wood R, Tsiamis A.: Factors determining the most efficient spray distribution for marine cloud brightening. Phil. Trans. R. Soc. A 372: 20140056. http://dx.doi.org/10.1098/rsta.2014.0056, 2014b.

Crooks, M., Connolly, P., and McFiggans, G.: A parameterisation for the co-condensation of semi-volatile organics into multiple aerosol particle modes, Geosci. Model Dev. Discuss., https://doi.org/10.5194/gmd-2017-123, in review, 2017.

Fountoukis, C., and Nenes A.: Continued development of a cloud droplet formation parameterization for global climate models, *J. Geophys. Res.*, 110, D11212, doi:10.1029/2004JD005591, 2005.

Ghan, S. J., Abdul-Razzak, H., Nenes, A., Ming, Y., Xiaohong, L., Ovchinnikov, M., Shipway, B., Meskhidze, N., Xu, J., and Shi, X.: Droplet nucleation: Physically-based parameterizations and comparative evaluation, J. Adv. Model. Earth Syst., 3, M10001, doi:10.1029/2011MS000074, 2011.

Jimenez, J. L., Canagaratna, M. R., Donahue, N. M., Prevot, A. S. H., Zhang, Q., Kroll, J. H., DeCarlo, P. F., Allan, J. D., Coe, H., Ng, N. L., Aiken, A. C., Docherty, K. S., Ulbrich, I. M., Grieshop, A. P., Robinson, A. L., Duplissy, J., Smith, J. D., Wilson, K. R., Lanz, V. A., Hueglin, C., Sun, Y. L., Tian, J., Laaksonen, a, Raatikainen, T., Rautiainen, J., Vaattovaara, P., Ehn, M., Kulmala, M., Tomlinson, J. M., Collins, D. R., Cubison, M. J., Dunlea, E. J., Huffman, J. A., Onasch, T. B., Alfarra, M. R., Williams, P. I., Bower, K., Kondo, Y., Schneider, J., Drewnick, F., Borrmann, S., Weimer, S., Demerjian, K.,Salcedo, D., Cottrell, L., Griffin, R., Takami, A., Miyoshi, T., Hatakeyama, S., Shimono, A., Sun, J. Y., Zhang, Y. M., Dzepina, K., Kimmel, J. R., Sueper, D., Jayne, J. T., Herndon, S. C., Trimborn, A. M., Williams, L. R., Wood, E. C., Middlebrook, A. M., Kolb, C. E., Baltensperger, U., and Worsnop, D. R.: Evolution of organic aerosols in the atmosphere, Science, 326, 1525–1529, doi:10.1126/science.1180353, 2009.

Simpson, E., Connolly, P., and McFiggans, G.: An investigation into the performance of four cloud droplet activation parameterisations, Geosci. Model Dev., 7, 1535-1542, https://doi.org/10.5194/gmd-7-1535-2014, 2014.

---

## Referee Comment (RC2) · Anonymous Referee #2 · 23 Jun 2018

In this paper the authors discuss results from a box-model sensitivity study comparing changes in parameterized cloud droplet numbers to a representation of semi-volatile organic partitioning.

Before consideration for publication, there are a number of issues that need addressing that are raised below.

Having read the original paper on the MATRIX-VBS model, I couldn't tell whether, since you are prescribing aqueous solubility, water is explicitly included in the organic partitioning simulations? If not, there is an inconsistency between prescribing a 'kappa' value [which will not stay constant unless 100% solubility is assumed] and assuming completely 'dry' partitioning unless all organics are thus actually assumed to have zero aqueous solubility. If so, please describe how you have accounted for varying solubility,

[Figure]

and presumably molecular weight, in the new partitioning simulations.

With varying solubility per bin, how would you then account for the influence of one VBS bin on the other in-line with mixing thermodynamics?

If you do include water in the partitioning simulations, how might you account for this in equilibrium partitioning at 100%RH?

How does the fixed linear change with solubility map to the VBS source VOCs used in a host model? For example, the use of experimentally determined RH variation in Kappa from isoprene and monoterpene SOA experiments has been shown to have significant impacts on two state-of-the-art climate model forcing estimates [Microphysical explanation of the RH dependent water affinity of biogenic organic aerosol and its importance for climate N. Rastak et al. https://doi.org/10.1002/2017GL073056]

It would seem the crux of the conclusions rests on the above process description and how the ARG parameterization takes that information to predict cloud droplet number. ARG would not capture partitioning through the humidity life-cycle, so please elaborate on the link between partitioning within the VBS model at any given RH to feeding parameters into ARG.

The title is certainly a question worth asking. However I wonder whether results from a model sensitivity study that, whilst interesting, rests on a framework that does not apparently capture process level phenomena which would influence results can be used to deliver an answer. Starting with responses to the questions above, I would suggest the following statement requires re-phrasing: 'We expect that implementing the improved box model in the global scale that includes a two moment cloud microphysical scheme (Morrison and Gettelman, 2008; Gettelman and Morrison, 2015) would more accurately represent aerosol-cloud interactions, which will be our focus on a follow up study. Thus it would offer us valuable insights on how the addition of organic partitioning would change cloud activation in the global atmosphere and its implications for climate.' There is no indication that process representation within this study

has improved on any previous. Conflicting implications on process combinations re-strict this evaluation. To more accurately represent aerosol- cloud interactions through an attempt to account for organic solubility and volatility, more detail is needed before publication. The alternative, of course, is to not present this as an improved represen-tation but deliver it as an existing model sensitivity study which might be better suited to publication in Geoscientific Model Development.

---

## Author Comment (AC1) · 22 Jul 2018

Reply to the 2nd review of "Can Semi-Volatile Organic Aerosols Lead to Less Cloud Particles?"

We would like to thank the reviewer for their efforts in evaluating our submission and providing constructive comments. Please find below our answers to all points raised. The original reviewer's comments are in black font, and our replies are in blue. Changes in the text are *italicized*.

In this paper the authors discuss results from a box-model sensitivity study comparing changes in parameterized cloud droplet numbers to a representation of semi-volatile organic partitioning. Before consideration for publication, there are a number of issues that need addressing that are raised below.

1. Having read the original paper on the MATRIX-VBS model, I couldn't tell whether, since you are prescribing aqueous solubility, water is explicitly included in the organic partitioning simulations? If not, there is an inconsistency between prescribing a 'kappa' value [which will not stay constant unless 100% solubility is assumed] and assuming completely 'dry' partitioning unless all organics are thus actually assumed to have zero aqueous solubility. If so, please describe how you have accounted for varying solubility, and presumably molecular weight, in the new partitioning simulations. With varying solubility per bin, how would you then account for the influence of one VBS bin on the other in-line with mixing thermodynamics? If you do include water in the partitioning simulations, how might you account for this in equilibrium partitioning at 100%RH? How does the fixed linear change with solubility map to the VBS source VOCs used in a host model? For example, the use of experimentally determined RH variation in Kappa from isoprene and monoterpene SOA experiments has been shown to have significant impacts on two state-of-the-art climate model forcing estimates [Microphysical explanation of the RH dependent water affinity of biogenic organic aerosol and its importance for climate N. Rastak et al. https://doi.org/10.1002/2017GL073056]

All aerosol populations in the simulations (including organic-containing aerosol populations) include water as a component, however, water is not considered in the partitioning of organics. They are also separate processes: one is to get organics on the aerosols due to partitioning, another is to grow aerosols with water. We do not take any Henry solubility into account when calculating partitioning, we use the Pankow parameterization (Pankow, 1994) which does not take into account water. For the cloud parameterization we calculate kappa from the chemical composition of the populations. This is not 100% consistent with each other, but it is the state of the art in global climate models. This would certainly be an interesting question to pursue, but is beyond the

scope of this study. As for kappa per VBS bin, we have a constant kappa per VBS bin listed in Table 1 of the manuscript, and the molecular weight of the VBS species are considered constant.

*To include this detail, we have included the following in line 85 of our text: "....we assumed a linear increase of solubility with decreasing volatility (Jimenez et al., 2009). Since we use Pankow type partitioning (Pankow, 1994), water is not considered in the partitioning process. In addition, we do not use different kappa/RH relationships per organic species, which was found to be important for biogenic SOA (Rastak et al., 2017)."*

4. It would seem the crux of the conclusions rests on the above process description and how the ARG parameterization takes that information to predict cloud droplet number. ARG would not capture partitioning through the humidity life-cycle, so please elaborate on the link between partitioning within the VBS model at any given RH to feeding parameters into ARG.

In the model, partitioning occurs before activation as a distinct, not synchronous, process. During activation when we feed information into the ARG scheme, there's no partitioning anymore. As we mentioned in the previous answer, water is not considered, so activated particles, which are essentially cloud droplets now, will not be part of any further partitioning. Since this is a box model, the activated particles are calculated as a diagnostic variable in the model, not a prognostic one. Our next paper will be a global model study, where this link can be explored further. Please also see response to comment 1 from Reviewer #1.

5. The title is certainly a question worth asking. However I wonder whether results from a model sensitivity study that, whilst interesting, rests on a framework that does not apparently capture process level phenomena which would influence results can be used to deliver an answer. Starting with responses to the questions above, I would suggest the following statement requires re-phrasing: 'We expect that implementing the improved box model in the global scale that includes a two moment cloud microphysical scheme (Morrison and Gettelman, 2008; Gettelman and Morrison, 2015) would more accurately represent aerosol-cloud interactions, which will be our focus on a follow up study. Thus it would offer us valuable insights on how the addition of organic partitioning would change cloud activation in the global atmosphere and its implications for climate.' There is no indication that process representation within this study has improved on any previous. Conflicting implications on process combinations restrict this evaluation.

This model captures process level phenomena for aerosol microphysics with consideration of nucleation, condensation, and coagulation, all of which affect the activated number concentration calculated by the model at any given time. The new model has in addition organic aerosol partitioning, a process previously missing from the original version of the model, adding an extra process that affects aerosol microphysics.   However, we will need to use the global model coupled with cloud microphysics to look at process level phenomena that would affect clouds following aerosol activation. We have also re-phrased the statement above with, "*We expect that implementing the improved box model in the global scale that includes a two moment cloud microphysical scheme (Morrison and Gettelman, 2008; Gettelman and Morrison, 2015) would more accurately represent aerosol-cloud interactions, which will be our focus on a follow up study.Thus it would offer us valuable insights on how the addition of process level phenomena in aerosol microphysics, as applied here for the organics partitioning, would affect cloud microphysics in the global atmosphere and its implications for climate.*" in the manuscript.

For the comment that "There is no indication that process representation within this study has improved on any previous. Conflicting implications on process combinations restrict this evaluation.", we would like to point out that this study isn't an evaluation of the old or the new model.  The partitioning process is quite uncertain and the VBS framework is heavily tuned against certain measurements. There are a lot of degrees of freedom in the system and it cannot be evaluated properly in any process-level study without a chamber simulation designed exactly for that purpose, which this paper is not trying to do. In order to test model skill for its climate implications, the evaluation should be performed on a global scale, which we are doing at the moment and plan to publish soon. We will evaluate whether this would improve the model or not, but it is important to note that partitioning affects microphysics in complex ways that are not easy to estimate without a model simulation, and that's what makes this study valuable.

6. To more accurately represent aerosol- cloud interactions through an attempt to account for organic solubility and volatility, more detail is needed before publication. The alternative, of course, is to not present this as an improved representation but deliver it as an existing model sensitivity study which might be better suited to publication in Geoscientific Model Development

We would like to emphasize that we do not study aerosol-cloud interactions here, only the initial phase of cloud formation. Again, we do not claim that the new model has improved in terms of results, but it is different from that of the original model. And as

mentioned in responses to the questions above, it has certainly improved in terms of processes.

References:

Pankow, J. F.: An absorption model of gas/particle partitioning of organic compounds in the atmosphere, Atmos. Environ., 28, 185–188, 1994.

Rastak, N., Pajunoja, A., Navarro, J. C. A., Ma, J., Song, M., Partridge, D. G., Kirkevag, A., Leong, Y., Hu, W. W., Taylor, N. F., Lambe, A., Cerully, K., Bougiatioti, A., Liu, P., Krejci, R., Petaja, T., Percival, C., Davidovits, P., Worsnop, D. R., Ekman, A. M. L., Nenes, A., Martin, S., Jimenez, J. L., Collins, D. R., Topping, D. O., Bertram, A. K., Zuend, A., Virtanen, A., and Riipinen, I.: Microphysical explanation of the RH-dependent water affinity of biogenic organic aerosol and its importance for climate, Geophys. Res. Lett., 44, 5167-5177, 10.1002/2017gl073056, 2017.

---

## Author Response (AR2)

Dear Editor,

We received your comments and suggestions for our discussion paper "Can Semi-Volatile Organic Aerosols Lead to Less Cloud Particles?" on ACPD. We thank you very much for your efforts in evaluating our submission.

Of the final referee's comment, we have modified this line in the Conclusions section, "*We would like to emphasize that our results do not imply that the Earth has less CCN than currently thought. Instead, they imply that if in a model semi-volatile organics will be simulated together with aerosol microphysics, a general decrease is to be expected.*", which now reads, "*We would like to emphasize that our results do not imply that the Earth has less CCN than currently thought. Instead, they imply that if in a model semi-volatile organics will be simulated together with aerosol microphysics, a general decrease is to be expected, assuming our model captures all relevant contributory processes.*" This change is highlighted in the updated revised manuscript.

We hereby resubmit the revised discussion paper to be considered for publication in *Atmospheric Chemistry and Physics.* We confirm that all authors listed on the manuscript concur with submission in its revised form. Should you have any remaining questions, we will be happy to address them.

Sincerely,

C. Y. Gao, S. E. Bauer and K. Tsigaridis

[revised manuscript text omitted]